# Aquaporin Channels in Skin Physiology and Aging Pathophysiology: Investigating Their Role in Skin Function and the Hallmarks of Aging

**DOI:** 10.3390/biology13110862

**Published:** 2024-10-24

**Authors:** Nazli Karimi, Vahid Ahmadi

**Affiliations:** 1Physiology Department, Medical Faculty, Hacettepe University, Ankara 06800, Turkey; 2Dermatology Department, Beytepe Murat Erdi Eker State Hospital, Ankara 06800, Turkey

**Keywords:** skin physiology, aging pathology, aquaporins

## Abstract

The skin, the body’s largest organ, is vital for protecting and maintaining the body’s health. Aging induces structural and functional changes, which lead to a reduction in skin resilience. This review investigates aquaporins, crucial proteins primarily known for water transport which are now recognized for their other vital roles in skin besides hydration such as barrier integrity and immune function. These channels achieve this primarily through the transport of important molecules and the activation of key molecular pathways for regulating cell function. Disruptions to aquaporin function are associated with various dermatological disorders. Understanding the mechanisms through which aquaporins function can enhance our knowledge of skin physiology and identify novel therapeutic targets for promoting healthy aging. This analysis provides an overview of how aquaporins can be utilized to manage skin aging and aging-related conditions.

## 1. Introduction

The skin serves as a primary defense mechanism against infection and other injuries, regulates temperature, produces vitamins, and enables sensory interaction with the environment [1]. Furthermore, the skin preserves fluid and electrolyte balance, supports blood volume, and contains a pivotal immune system that coordinates our interaction with the external world [2]. Its appearance can indicate health or disease, leading to significant psychological impacts [1]. In today’s world, there is a heightened recognition of skin health as a significant indicator of overall wellbeing, resulting in an increased demand for anti-aging products and treatments. As individuals age, the skin undergoes structural and functional imbalances that impair its functions [3]. Studies show that targeting aging processes, such as boosting skin health, offers greater economic benefits than addressing specific diseases, with significant financial gains from slowing the aging process [4]. Aquaporins (AQPs), a family of transmembrane channel proteins, play a crucial role in the skin by facilitating the rapid transport of water and necessary small molecules between various cells throughout the body [5]. Eight AQPs (AQP 1, 3, 5, 7, 9, and 10) [6], AQP11 [7], and AQP8 [8] are found in different cell types in the skin. These proteins are involved in various skin functions, including proper hydration, supporting the skin barrier’s integrity, and promoting skin health [5,6]. Recent research suggests that changes in the expression of AQPs are associated with skin aging, indicating that aquaporins may play a regulatory role in the molecular process skin aging [5,8,9,10]. As a result, this review focuses on exploring the role of aquaporins in skin physiology, emphasizing their modulatory function on the hallmarks of aging and their potential as therapeutic targets.

## 2. Hallmarks of Aging

Aging is a complex process characterized by interconnected molecular and cellular mechanisms. Phenotypically, it involves a gradual decline in cellular function and systemic tissue deterioration, leading to increased vulnerability to age-related diseases [11]. Human aging, characterized by a deterioration in functions and quality of life [12], presents significant challenges as life spans increase while health spans lag behind [13], resulting in healthcare and socioeconomic burdens [4,14]. Therefore, studying the biology of aging and developing interventions to extend healthy lifetimes are key to easing the global pension burden and promoting healthy aging.

In 2013, nine core hallmarks of aging were identified, including deoxyribonucleic acid (DNA) instability, telomere shortening, epigenetic changes, a loss of protein balance, disrupted nutrient sensing, mitochondrial dysfunction, cellular senescence, stem cell depletion, and altered intercellular communication [15]. Recently, the hallmarks of aging have been updated to include impaired macroautophagy, chronic inflammation, and dysbiosis. An age-related decline in autophagy is a key factor in reduced organelle turnover, making it a significant new hallmark of aging [16]. It must be considered that all aging hallmarks are interconnected and influence one another, rather than operating independently.

## 3. Skin Aging

The skin, as the body’s primary interface with the environment [17], functions as both a protective barrier [18] and a visible marker of aging [19]. The skin barrier consists of four key components: the physical barrier, chemical barrier, immune barrier, and microecological barrier [20]. Skin aging not only impacts its appearance and function but also has wider implications for systemic aging. As a protective barrier, skin is reinforced by both innate and adaptive immune responses [21]. The regulation of this barrier involves a complex network of molecular and immunological signaling pathways [22]. One notable change that occurs with aging is the weakening of the skin barrier’s function [23]. Recent studies have shown that epidermal dysfunction contributes to chronic, low-grade systemic inflammation through the whole body which is associated with the onset of age-related systemic disorders [24]. Although other studies demonstrate that chronic inflammatory skin conditions, such as psoriasis and atopic dermatitis (AD), are frequently associated with a higher risk of obesity, prediabetes, type 2 diabetes, hypercholesterolemia, and hypertension, all of which contribute to the progression of age-related chronic diseases [25], Ye et al. conducted a pilot study to assess whether improving epidermal barrier function in older adults could reduce circulating pro-inflammatory cytokines. They found that the topical application of a barrier-repair emollient led to a reduction in IL-1β and IL-6 levels, bringing them to levels similar to those of younger controls. TNFα levels also decreased in the treated elderly participants, with no significant difference compared to the young control group [26]. These findings emphasize that impaired skin barrier function can trigger systemic inflammation. As the largest organ, the skin experiences both intrinsic and extrinsic aging, triggering structural and cellular changes with the accumulation of senescent cells. These cells can accelerate dysfunction in surrounding tissues and contribute to aging in other organs [27]. The extent of skin aging may serve as a predictor of life expectancy [28]. With age, the skin develops visible signs such as wrinkles, pigmentation changes, rough texture, dryness, and sagging. Endogenous (intrinsic or chronological) skin aging is driven by genetic factors such as genomic instability, cellular senescence, and telomere shortening [27], while exogenous aging is influenced by environmental factors like ultraviolet (UV) radiation, pollution, and lifestyle choices [29]. UV radiation accelerates skin aging in a dose-dependent manner by inducing oxidative stress and inflammation, which are both important hallmarks of skin damage [28,30].

Skin aging involves a decline in collagen content, reduced skin thickness, dryness, and wrinkle formation. This process is driven by mechanisms including the free radical theory, inflamm-aging, photoaging theory, and metabolic theory [28]. Primary signs of aging include thinning of the stratum corneum, reduced epidermal enzyme activity, and impaired repair capacity, with recent evidence linking these changes to decreased water content, reduced moisturizing capacity, and a loss of skin elasticity (Figure 1) [31]. Understanding the mechanisms of age-related skin dysfunction could lead to therapies that improve skin health and may also slow aging-related disorders in other organs.

## 4. Overview of Aquaporins

The existence of specialized water-conducting channels was first indicated by Benga’s group in the 1980s, and later, Peter Agre’s group identified and confirmed the water-conducting properties of the same channel [32], initially named channel-forming integral protein of 28 kDa (CHIP28) and later renamed AQP1. The detailed history of this discovery is discussed in Kuchel’s article [33]. AQPs are essential integral membrane proteins that facilitate the transmembrane transport of water. These structures increase the permeability of cellular plasma membranes to water by 5–50 times compared to the lipid bilayers through which water normally passes [34]. The regulation of water flux into and out of cells is a ley biological process essential for sustaining life, influencing not only cellular function but also the operation of multiple organ systems and the maintenance of systemic water homeostasis [35]. AQPs are found across diverse life forms and share a common structure of six transmembrane alpha helices forming a central water-conducting channel. Four AQP monomers assemble into functional tetramers. Initially discovered as water channels, subsequent studies have revealed that aquaporins also facilitate the diffusion of various solutes, including neutral and charged molecules, small and bulky molecules, and even gasses [36]. Additionally, AQPs regulate various cellular processes by controlling osmotic water flow during cell growth, energy metabolism, cell migration, adhesion, and proliferation [37]. Unexpected roles include influencing tumor angiogenesis, neural signal transduction, and inflammation [38]. Their functions are regulated by posttranslational modifications such as phosphorylation, ubiquitination, and glycosylation. Understanding these mechanisms is vital for developing new therapeutic targets and diagnostic biomarkers [39].

At present, 13 recognized isoforms of AQPs exist in mammals (AQP0–AQP12), though the precise roles of some remain to be determined [5]. Ishibashi et al. classified the AQPs family into three subfamilies based on unique primary sequences [40]. The first category, known as orthodox or classical AQPs, includes highly selective water channels such as AQP0, AQP1, AQP2, AQP4, AQP5, AQP6, and AQP8 [5,40,41]. Aquaglyceroporins comprise the second category, which—besides transporting water—are permeable to small non-polar molecules like glycerol and urea. This group includes AQP3, AQP7, AQP9, and AQP10 [41,42]. The third category, unorthodox AQPs, includes AQP11 and AQP12. These are distinct due to their atypical structure, with less than 20% sequence similarity to others. Initially believed to transport other molecules, these unorthodox AQPs were later found to also transport water, leading to their classification as atypical aquaporins or supra-aquaporins [43]. Also, superaquaporins are uniquely situated in the membranes of intracellular organelles. It is proposed that AQP11 and AQP12 are involved in the transport of water and glycerol within cells, participating in regulating organelle volume and homeostasis [44,45,46]. Also, in recent years, increased interest in oxidative stress has identified several AQP isoforms (AQP1, AQP3, AQP5, AQP8, AQP9, AQP11) as peroxiporins, facilitating hydrogen peroxide (H_2_O_2_) permeation [47,48]. This ability to transport H_2_O_2_ highlights the role of peroxiporins as key regulators in oxidative processes and cell-to-cell communication, with significant implications for understanding and treating diseases associated with oxidative stress and inflammation [47]. An increasing number of studies demonstrate the diverse activity of AQPs in fundamental processes and also signaling pathways. Altered AQP expression has been associated with various pathophysiological conditions [49]. In the context of aging, reduced AQP expression in skin cells and tissues points to their responsibility in maintaining skin homeostasis, making AQPs promising candidates for developing cosmetics and pharmaceuticals aimed at attenuating skin aging [50].

Another key aspect is that AQPs are regulated at three primary levels, transcriptional, translational, and post-translational [51], all of which are essential for maintaining their proper function in skin physiology. Since AQPs are typically expressed constitutively in plasma membranes, their functional regulation primarily occurs at the transcriptional level [52]. In transcriptional regulation, transcription factors such as Nuclear Factor kappa B (NF-κB) and Hypoxia-Inducible Factor (HIF) play crucial roles in controlling AQP gene expression. NF-κB is activated by inflammatory signals and can change AQP expression during inflammatory conditions [53], while HIF regulates AQPs under low oxygen conditions, adjusting their levels during hypoxia [54,55,56]. In translational regulation, microRNAs (miRNAs) modulate AQP expression by binding to AQP mRNA transcripts, inhibiting translation and thereby controlling protein synthesis; this mechanism allows for rapid adjustments of AQP levels in response to cellular signals [57,58,59]. Post-translational regulation involves modifications such as phosphorylation and glycosylation, which significantly affect AQP function [39]. Phosphorylation can alter water permeability and AQP localization, while glycosylation impacts their stability and trafficking to the plasma membrane [60], ensuring proper skin barrier function and hydration.

## 5. Skin and Aquaporins

### 5.1. Structure and Function of the Skin

The skin consists of three distinct layers: the epidermis, dermis, and hypodermis (subcutaneous tissue) [9,30]. The epidermis, the outermost layer, is composed of five sublayers: the basal layer, spinous layer, granular layer, translucent layer, and the stratum corneum (SC). The SC, consisting of mature keratinocytes, is mainly known for facilitating the skin’s barrier function [61,62]. Beneath the epidermis lies the dermis, which contains collagen, elastin, and fibroblasts, providing structural support. The epidermal basement membrane connects the dermis and epidermis, helping maintain skin integrity and stability. The epidermis consists of various cell types, including keratinocytes, melanocytes, Langerhans cells, and Merkel cells [9]. Keratinocytes, the primary cell type in the epidermis, along with lipids and other components, form a protective barrier against environmental elements, prevent moisture loss, and synthesize keratin for structural support [63,64]. Epidermal stem cells (ESCs) in the basal layer self-renew and differentiate, contributing to the continuous cycle of skin regeneration through proliferation, differentiation, and apoptosis [65]. Unlike the epidermis, which is dense with keratinocytes, the dermis is primarily composed of the acellular extracellular matrix (ECM). Collagen fibers, constituting 75% of the skin’s dry weight, provide tensile strength and elasticity, while elastic fibers allow the skin to return to its original shape after stretching. The ECM also contains proteoglycans (PGs) and glycosaminoglycans (GAGs), which absorb significant amounts of water and regulate the dermis’s water-binding capacity and compressibility. Fibroblasts, derived from mesenchymal cells, are central in synthesizing and degrading ECM proteins. Other dermal cells include immune cells such as histiocytes, mast cells, dermal dendrocytes, and endothelial cells [66]. In aging skin, collagen production decreases while its degradation increases, leading to a reduction in overall collagen content [3,67]. Beneath the dermal–epidermal junction is the hypodermis or subcutaneous layer, composed mainly of fat and connective tissue [61]. This adipose tissue is divided into subcutaneous adipose tissue and dermal white adipose tissue (dWAT), the latter surrounds the proximal portions of hair follicles and displays unique characteristics in terms of cell renewal, precursor cell populations, and differentiation patterns, compared to fat in other areas of the body [8]. Aging leads to the thinning of the hypodermis, which serves as a major stem cell niche and a source of hormones and adipokines. This thinning contributes to skin sagging and has a detrimental effect on the dermal microenvironment and overall skin function [68]. Adipose stem cells (ASCs) from dWAT are considered to be an important stem cell population for dermal regeneration and play a critical role in epithelialization [69,70]. Research also revealed that the adipocyte lineage within dWAT is involved in the healing of acute skin wounds [71]. Skin health depends on the integrated action of various molecules and cells to regulate epidermal function and repair damage from various causes. These processes have excessive importance for maintaining the skin’s physiological functions, involving barrier integrity and regeneration [72]. Disruptions in these pathways resulted in hypodermal thinning and reduced ASCs activity which can impair skin repair and contribute to aging-related changes.

### 5.2. Roles of Aquaporins in Maintaining Skin Function

#### 5.2.1. Distribution of Aquaporins in Skin

AQP expression has been detected in various skin structures [73]. AQP1 and AQP3 are distributed throughout both the dermis and epidermis [74]. AQP5 is localized in the eccrine sweat glands. AQP7 is primarily found in the hypodermis, with additional expression in dermal and epidermal dendritic cells (DCs). AQP8 is present in dermal fibroblasts, while AQP9 and AQP10 are expressed in the epidermis cells [50]. AQP11 is mainly located in hypodermal fat [46].

#### 5.2.2. Functions of AQP1 and AQP3 in Skin: Water Transport, Cell Migration, Skin Hydration, and Barrier Maintenance

AQP1 has been identified in the endothelial cells of dermal capillaries [75] and in cultured melanocytes and fibroblasts [2]. AQP1 plays a vital role in osmotic water transport across cell membranes, particularly in epithelial and endothelial tissues [75]. While AQP1 mainly facilitates water transport, it also acts as a cation channel [76], gated by cyclic guanosine monophosphate (cGMP) and indirectly regulated by cyclic adenosine monophosphate (cAMP) [77]. AQPs, besides having essential roles in water transport, are essential for promoting cell migration in the skin. During migration, cells form membrane protrusions, such as lamellipodia and ruffles, at the leading edge, which are necessary for movement. These structures require rapid ion flux and cell volume changes, processes facilitated by AQPs [78]. Without AQPs, the limited water permeability of the cell membrane slows down the changes required for the cell migration process which is needed for processes like wound healing, where rapid cell migration is essential [79,80]. For example, the absence of AQP1 impairs endothelial cell migration, and the lack of AQP3 slows down keratinocyte migration, providing AQPs with a role in these processes [80]. The influx of water through AQP1 or AQP3 is thought to provide the hydraulic pressure necessary for extending cellular processes during movement [74]. AQP3, predominantly expressed in the keratinocytes of the stratum basale (SB) and stratum spinosum (SS), serves as a core channel for transporting water, glycerol, and H_2_O_2_ [81]. Its higher expression in the basal and spinous layers of the epidermis contributes significantly to their increased water content (75%) compared to the SC (10–15%) [5,82]. Data from AQP3 knockout mice mark the activity of AQP3 in vivo, as these mice exhibit reduced water retention, decreased skin elasticity, and impaired recovery of water permeability after barrier disruption. Indeed AQP3 knockout mice exhibit significantly impaired barrier homeostasis [83]. On the other hand, an enhanced expression of AQP3 in the epidermis, regulated by the keratin-1 promoter, speeds up the repair of the epidermal barrier [84].

Moreover, glycerol supplementation in AQP3-null mice effectively restored SC hydration and improved skin elasticity, with both topical and systemic treatments normalizing SC glycerol levels, pinpoint the unique role of AQP3 in glycerol transport [85]. In the asebia mouse model, where reduced sebum production led to lower epidermal glycerol levels and impaired SC hydration, glycerol treatment reversed hyperkeratosis. Re-expression of AQP3 in keratinocytes corrected the skin abnormalities in AQP3-deficient mice, further proving the essential role of AQP3-mediated glycerol transport in maintaining skin hydration and function [85,86]. Overall, the elevated expression of AQP3 facilitates water transport, and maintains optimal moisture levels throughout the epidermis [87]. The moisturizing capacity of the skin is closely linked to AQP3′s function in balancing water. Skin hydration is commonly assessed by measuring tissue water content in the SC [88], dermis, and deeper layers [89], as well as by evaluating transepidermal water loss (TEWL) [90]. Moisturizing factors such as collagen [91], ceramides [92], and hyaluronic acid [93] also regulate skin water content, with AQP3 playing an essential role in water retention [9]. AQP3-knockout mice exhibit dry skin with decreased elasticity, underlying its role in skin hydration [84]. Aberrant AQP3 expression is associated with atopic dermatitis and psoriasis, affecting skin hydration [94]. In addition, in the human epidermis and cultured keratinocytes, AQP3 supports the absorption of urea, an essential metabolite that significantly enhances the hydration of the SC [95]. AQP3 expression significantly declines in the non-sun-exposed skin of individuals over 60 compared to those under 45, proving its importance in intrinsic skin aging [96]. Studies also show reduced AQP3 levels in older mice compared to younger ones [97]. External factors like UV radiation and reactive oxygen species can also reduce AQP3 levels in skin keratinocytes [98]. This reduction is observed in both sun-exposed and non-sun-exposed skin, linking AQP3 to both intrinsic and extrinsic aging. Given AQP3′s primary role in skin hydration, its downregulation may significantly contribute to the dryness seen in aging skin [31]. AQP3′s involvement in skin hydration is further enhanced by its circadian rhythm of expression, with hydration levels fluctuating in sync with AQP3 cycles [99]. Recent research revealed AQP3′s vital function in modulating keratinocyte proliferation and differentiation [9]. During differentiation, AQP3 associates with phospholipase D2 (PLD2) in caveolin-enriched microdomains of the plasma membrane. AQP3 facilitates glycerol import, which PLD2 uses to generate phosphatidylglycerol (PG), a signaling lipid essential for early keratinocyte differentiation. Disruption of this pathway has been shown to inhibit keratinocyte proliferation while increasing differentiation [84].

Impaired keratinocyte proliferation can lead to epidermal thinning, enhancing fragility. Hara-Chikuma and colleagues have demonstrated that AQP3 supports cell proliferation in both mouse and human keratinocytes [100,101,102]. Reducing AQP3 levels in human keratinocytes leads to decreased cell proliferation, while AQP3 deficiency in knockout mice results in lower cellular ATP levels and impaired cell growth [82]. Additionally, the co-expression of AQP3 with reporter genes, where luciferase expression is controlled by the promoters for keratin 1 or involucrin (markers of early and intermediate keratinocyte differentiation, respectively), leads to increased luciferase activity under both basal and differentiative conditions [103]. Moreover, the re-expression of AQP3 in AQP3-deficient keratinocytes enhances the expression of keratinocyte differentiation markers, further suggesting a role for this channel in the differentiation process [86]. In addition to its presence in keratinocytes, AQP3 is expressed in cultured human skin fibroblasts, which has a critical function in wound healing mechanisms by migrating from surrounding tissues adjacent to the wound. Research has shown that epidermal growth factor (EGF) activates the epidermal growth factor receptor (EGFR), thereby stimulating cell migration. Notably, EGF has been shown to upregulate AQP3 expression, further improving fibroblast migration during wound healing [104]. This suggests a double role for AQP3 in both keratinocyte differentiation and fibroblast-mediated wound repair.

#### 5.2.3. Role of AQP5 in Keratinocyte Regulation and Sweat Gland Function

AQP5 overexpression in HaCaT cells has been shown to enhance keratinocyte proliferation and dedifferentiation without impacting apoptosis, emphasizing its regulatory role in maintaining keratinocyte balance. Significantly, AQP5 expression declines with age, which adversely affects epidermal stem cell functions and can contribute to age-related skin disorders [10]. Also, AQP5 is critical in the permeability of the apical membrane of sweat gland secretory cells, promoting efficient sweat secretion [5].

#### 5.2.4. Roles of AQP3, AQP 5, AQP7, and AQP8 in Immune Function and Oxidative Stress Protection in the Skin

AQP3 is critical for several immune processes in the skin. It facilitates H_2_O_2_ influx into CD8+ T cells, which is essential for endocytosis, and also participates in phagocytosis in macrophages and micropinocytosis in dendritic cells. Moreover, AQP3 is vital for the migration of T cells and macrophages, influencing their physiological role in immune defense [105,106,107]. Additionally, AQP3 regulates inflammatory responses by promoting IL-6, pro-IL-1β, and TNFα transcription via TLR4 engagement upon LPS stimulation, and it plays a role in NLRP3 inflammasome activation [48,108]. This AQP is upregulated during immune activation, enabling the rapid cell volume changes necessary for IL-1β secretion (81). AQP5 is involved in the migration of neutrophils and plays a role in the priming and proliferation of dendritic cells [109,110]. This regulation of cell mobility and immune function is essential for the skin’s response to inflammation and injury [109]. In the skin, DCs such as Langerhans cells and dermal dendritic cells contribute significantly to initiating the immune response by capturing antigens and migrating to regional draining lymph nodes (LNs) to activate naive T cells. AQP7, expressed in dendritic cells, facilitates antigen uptake and endocytosis, contributing to immune surveillance and responses in the skin. AQP7 also participates in the migration of dendritic cells towards chemoattractants during immune responses [111,112]. Research further indicates AQP7’s central function in allergy induction and immune surveillance in the skin and other tissues where DCs are present [74]. Based on the protein ATLAS database, AQP8 is predominantly expressed in skin fibroblasts, where it may act as a dominant factor in protecting against H_2_O_2_-induced oxidative damage. By mitigating oxidative stress, AQP8 potentially shields human dermal fibroblasts from aging-related damage [50].

#### 5.2.5. Role of AQP9 in Keratinocyte Differentiation and Immune Response Regulation

AQP9, a water/glycerol channel, is expressed at the messenger ribonucleic acid (mRNA) level in differentiated normal human epidermal keratinocytes (HEK) [113]. Retinoic acid, known for stimulating keratinocyte differentiation, increases AQP3 expression but downregulates AQP9, indicating distinct regulatory mechanisms for these two AQPs. Vitamin D3 (Vit D3), another differentiation agent, significantly elevates AQP9 levels, suggesting AQP9′s involvement in highly differentiated keratinocytes. In the human epidermis, AQP9 co-localizes with occludin, a tight junction marker, in the upper stratum granulosum (SG) [114]. AQP9 is also present in various immune cells, including neutrophils, where it holds a primary position in the skin’s immune response. It is required for sensitization during cutaneous acquired immune reactions by regulating neutrophil function. This sensitization is particularly important in allergic diseases. One study shows that AQP9 deficiency impairs neutrophil sensitization, reducing their recruitment and weakening the immune response [115]. AQP9 is also present at the lamellipodium edges of leukocytes, regulating cell shape changes and migration in response to chemokine gradients [116]. As mentioned, it is crucial for neutrophil and T cell activation, as its upregulation during infections promotes the cell shape changes necessary for effective immune responses [115,117,118]. AQP9 plays a key role in infection control, particularly in conditions such as sepsis [118,119]. Alterations in these aquaporins can weaken the skin’s defense mechanisms, increasing susceptibility to various bacterial viral or fungal infections.

#### 5.2.6. Roles of AQP10 and AQP11 in Skin Barrier Function and Lipid Metabolism

AQP10, another aquaglyceroporin found in epidermal keratinocytes, has been localized to the SC of the human epidermis in vivo. Due to its presence in the SC, AQP10 is thought to play a similar role to AQP3, potentially contributing to the skin’s barrier function [74]. AQP 3, 7, 9, 10, and 11 are expressed in adipocytes and are involved in lipid metabolism [5]. Knocking down AQP10 in human differentiated adipocytes resulted in a 50% decrease in both glycerol and osmotic water permeability [120]. AQP11 is primarily found in mature adipocytes in omental and hypodermal fat, where it is localized to the endoplasmic reticulum (ER) membrane near lipid droplets. This positioning allows AQP11 to facilitate glycerol mobilization for triacylglycerol synthesis under normal conditions [45]. A summary of aquaporin functions in the skin is provided in Table 1.

## 6. Key Hallmarks of Aging in Skin and the Role of Aquaporins

### 6.1. Mitochondrial Dysfunction

An excess of reactive oxygen species (ROS) and a reduction in nitric oxide (NO) bioavailability can deteriorate cellular function, which is closely associated with the processes of aging and the development of fragility [121]. Beyond their primary role as the cell’s energy producers, mitochondria primarily contribute to triggering inflammation by releasing ROS and mitochondrial DNA (mtDNA), which activate inflammasomes and cytosolic DNA sensors. Also, mitochondrial dysfunction can lead to cell death by releasing caspase activators, nucleases, and other lethal enzymes from the intermembrane space, further contributing to tissue damage and age-related decline [122]. H_2_O_2_, a well-known ROS produced within mitochondria, was traditionally believed to diffuse freely across membranes. However, recent evidence reveals that the permeability of membranes, particularly mitochondrial membranes, restricts its diffusion. It is now understood that protein-facilitated pathways, probably involving AQPs, can aid in the transport of H_2_O_2_ due to its physicochemical similarity to water [121]. As mentioned earlier the mammalian AQP homologs currently identified as facilitating the passive diffusion of significant amounts of H_2_O_2_ include AQP1, AQP3, AQP5, AQP8, and AQP9 [123]. In 2010, AQP3 overexpression was shown to elevate intracellular H_2_O_2_ levels in mammalian cells, implicating AQP3 in various disease signaling pathways [124,125,126]. Both AQP3 and AQP8 facilitate the transport of nicotinamide adenine dinucleotide phosphate (NADPH) oxidase (NOX)-generated H_2_O_2_, which is critical for intracellular signaling, stress responses, and cell migration [124,127,128]. Moreover, AQP3 plays a key role in regulating T cell migration by enabling H_2_O_2_ transport and activating family of small GTPases (Rho) signaling [112]. Also, oxidative stress has been shown to modulate AQP3 expression, with reduced levels observed in both ultraviolet A (UVA)-exposed and aging skin, indicating its potential as a biomarker for skin aging [31,129,130,131]. The contribution of aquaporins to facilitating H_2_O_2_ transport across the mitochondrial membrane and their adjustment by H_2_O_2_ feature their imperative function in both regulating mitochondrial dynamics and responding to oxidative stress which significantly contributes to mitochondrial dysfunction and accelerates the aging process.

### 6.2. Cellular Senescence and Stem Cell Depletion

Cellular senescence serves a dual function in tissue repair and protecting against oncogenic damage by first inducing senescence and then recruiting immune cells to clear senescent cells. Failure in either of these steps increases the susceptibility to disease. Stem cell exhaustion occurs when cellular plasticity—essential for tissue repair—declines. Tissue repair relies on a modified microenvironment influenced by the secretion of cytokines, growth factors, and ECM modulators. This is partially driven by the senescence-associated secretory phenotype (SASP), which promotes cell dedifferentiation and plasticity across tissue compartments [16]. On the other hand, SASP can be characterized by the release of various cytokines and inflammatory mediators, contributing to inflamm-aging [15,16]. Conversely, pro-inflammatory signals can promote senescence as well [132]. Cellular senescence develops in response to both acute and chronic damage [133], with senescent cells gathering at changing rates across different tissues in humans [134]. This process predominantly affects fibroblasts, endothelial cells, and immune cells [135]. Senescent fibroblasts can accelerate age-related dysfunction in other skin cells and contribute to systemic inflammation [136]. Human dermal fibroblasts (HDFs), in particular, are highly sensitive to oxidative stress, where exposure to H_2_O_2_ causes a dose-dependent decrease in viability and an increase in ROS levels, indicating a severe oxidative challenge [137,138]. In vitro experiments demonstrated that AQP3 overexpression improves the viability of both young and senescent human dermal fibroblasts (HDF) and inhibits cellular senescence in HDFs [59]. Transcriptome sequencing studies have revealed a significant decline in AQP5 expression in ESCs with age [10]. ESCs, which reside in the basal layer of the epidermis, have the ability to self-renew and differentiate into various skin layers, supporting continuous proliferation, differentiation, and apoptosis [65]. However, with aging, ESC depletion occurs, impairing their ability to maintain tissue homeostasis and repair damaged skin [139].

### 6.3. Impaired Macroautophagy

Autophagy is fundamental for maintaining cellular homeostasis by breaking down and recycling damaged proteins and organelles. Its function allows cells to adapt to stress and maintain their activity over time. A decline in autophagic efficiency is linked to age-related degeneration and diseases, emphasizing its role in delaying aging and preserving cellular health [140]. While the precise relationship between autophagy and aging remains unclear, it is generally accepted that autophagy decreases with age, leading to the accumulation of cellular damage [141]. A study by Xie, H. et al. revealed that short-term UVA exposure upregulates AQP3 expression via JUN, enhancing autophagy by interacting with the death effector domain—containing protein (DEDD) and Beclin1. This promotes the removal of damaged components and reduces cellular senescence in skin fibroblasts. Conversely, prolonged UVA exposure reduces AQP3 expression, suppresses autophagy, and accelerates cellular aging [142]. Another study by Sebastian Yu et al. explored arsenic-induced autophagy in primary human keratinocytes. Interestingly, the arsenic-induced autophagy was suppressed when keratinocytes were pretreated with aquaporin inhibitors, such as Auphen or silver nitrate (AgNO_3_), or through RNA interference targeting AQP3. These findings suggest that aquaporin 3 plays a key role as a membrane channel in facilitating arsenic uptake and is involved in triggering autophagy in response to stress [143].

### 6.4. Dysbiosis

Dysbiosis is generally described as an alteration in the composition of resident commensal microbial communities compared to those typically found in healthy individuals [144]. This imbalance can adversely impact skin health due to the skin–gut axis, where gut tight junctions play a key role in maintaining skin homeostasis [145]. Modulating the microbiome through probiotics has been shown to improve gut and skin health, reduce skin inflammation, and support barrier function, thus restoring skin homeostasis [146,147,148]. Khmaladze et al. found that Lactobacillus reuteri DSM17938 improves skin barrier function by increasing AQP3 expression, providing relief for individuals with dry skin [149]. Similarly, Bifidobacterium animalis CGMCC25262 enhances skin health by upregulating AQP3 and other barrier-related genes, while reducing inflammation through the inhibition of extracellular signal-regulated kinase (ERK) phosphorylation [150]. These findings show that regulating AQP3 expression via probiotic interventions may help mitigate the effects of dysbiosis on skin aging and improve overall skin health.

### 6.5. Inflamm-Aging

Inflamm-aging refers to chronic, low-grade systemic inflammation that arises from disruptions which are related to various hallmarks of aging. This ongoing inflammation accelerates aging through the induction of oxidative stress and other damaging effects, which impair systems over time. Although the exact causes of inflamm-aging remain unclear, it is likely driven by immune system dysfunction and age-related changes [23]. Inflammaging and immunosenescence advance together, forming a reinforcing cycle. Chronic inflammation from inflamm-aging suppresses the adaptive immune system, accelerating immunosenescence. In turn, the weakened adaptive immunity leads to increased activation of the innate immune system, which perpetuates inflammation and sustains the cycle of inflammaging. This age-related deterioration results in a heightened vulnerability to various infections [151]. As the skin’s barrier weakens with aging, inflammation increases, causing a weakened pathogen response and leading to broader health impacts. It has been demonstrated that the decline in AQP3 expression with aging further impairs skin function, exacerbating inflammation [23]. Extensive research shows that AQPs are important for the key processes in the inflammatory response [53,152]. Inflammation has a permanent role in skin aging, often triggering cell death and disrupting the skin’s structure and immune function. Keratinocyte death, a well-known characteristic of inflammatory skin diseases, contributes to skin inflammation and aging [153]. Inflammation also drives cellular senescence. Senescent cells, while unable to divide, maintain a high metabolic activity and secrete SASP factors, including pro-inflammatory cytokines, chemokines, and proteases, disrupting the skin microenvironment [38]. Oxidative stress exacerbates this process by damaging DNA, releasing damage-associated molecular patterns (DAMPs), and also activating chronic inflammation [154]. This persistent inflammation, combined with necroptosis, promotes skin cell arrest, aging, and death. In addition, matrix metalloproteinases (MMPs), which are upregulated during inflammation, degrade skin collagen, further contributing to structural damage and aging [155,156]. As mentioned earlier, oxidative stress is closely associated with skin inflammation, and age-related inflammatory processes may arise from reduced levels of anti-inflammatory cytokines such as interleukin (IL)-10 [157]. A recent study revealed that exposure to H_2_O_2_ reduces IL-10 gene expression in dermal fibroblasts, which may aggravate inflammation and accelerate aging. Interestingly, the upregulation of AQP8 in response to H_2_O_2_ acts as a protective mechanism alleviating its pro-aging effects [50]. Notably, Yang et al. demonstrated an inverse relationship between SC hydration and both inflammation and serum tumor necrosis factor-alpha (TNFα) levels [158]. As discussed in detail, each hallmark of aging which contributes to a decline in the skin’s protective functions, impairing its ability to respond to pathogens effectively and thereby increasing the risk of systemic inflammation. AQPs’ involvement in the hallmarks of skin aging is summarized in Table 2.

## 7. Therapeutic Implications of Aquaporins

AQPs play outstanding roles as drug targets in conditions characterized by disrupted water homeostasis, such as brain edema, cancer metastasis, and inflammation. However, the advancement of aquaporin-targeted drug development has been limited. Although aquaporins are recognized as validated drug targets, it remains uncertain whether their pores are inherently resistant to pharmacological intervention or if they can be effectively blocked by small molecules. An alternative strategy for drug development involves targeting the regulatory mechanisms that control AQPs functions, which have been discussed in detail [159]. Various molecules, including metal compounds, small molecules, and biologics, have been shown to modulate aquaglyceroporin function. However, their mechanisms remain unclear, and their pharmacological efficacy in disease models is still limited. Also, further research is needed to improve selectivity and reduce side effects and toxicity [160]. Uncovering specific modulators can help clarify whether the effects observed are aquaporin-dependent or not [159]. As mentioned earlier, AQPs are essential for regulating water transport in the skin, and age-related decreases in these proteins lead to impaired hydration. Research shows that stimulating AQP biosynthesis, mostly AQP3, may boost skin hydration by modulating proteins such as cluster of differentiation 44 (CD44), claudin-1, and filaggrin, which are central to moisture maintenance. Therefore, targeting AQP3 upregulation presents a promising strategy for improving skin hydration in aging populations [161]. Several compounds, including all-trans retinoic acid (ATRA) [98], chrysin [129], and glycolic acid [162], have been demonstrated to effectively upregulate AQP3 expression in human keratinocytes. By enhancing AQP3 levels, these compounds help restore barrier function weakened by UV radiation, further supporting skin hydration and overall skin health. In addition, a clinical study revealed that gallic acid-binding flavan-3-ol, an oligomeric proanthocyanidin from red wine extract, enhanced skin moisture in the stratum corneum by increasing AQP3 expression level in these cell types [163].

One study explored the suppression of miR-551b-3p by long non-coding RNA plasmacytoma variant translocation 1 (LncRNA PVT1), which increased AQP3 expression and inactivated the ERK/p38 MAPK pathway. This mechanism inhibited the senescence of HDFs, suggesting an encouraging therapeutic approach to delay skin photoaging [59]. Additionally, activation of proliferator-activated receptor gamma (PPARγ) has been shown to significantly elevate AQP3 mRNA levels in cultured human keratinocytes (CHKs) in a dose- and time-dependent manner, leading to increased AQP3 protein and enhanced glycerol uptake. Topical application of ciglitazone, a peroxisome PPARγ activator, also elevated AQP3 expression in mouse skin, and similar effects were observed with other nuclear hormone receptors like liver X receptors (LXR), retinoic acid receptors (RAR), and retinoid X receptors (RXR). PPARs were also associated with the Histone Deacetylase (HDAC) inhibitor-induced increase in AQP3, as PPAR antagonists blocked this effect [164]. These findings highlight the therapeutic potential of targeting PPARγ and related pathways to enhance AQP3 expression, improve skin hydration, and combat skin aging. Supporting this, a recent study by Filatov et al. demonstrated that Aloe barbadensis extract and trimethylglycine significantly increased AQP3 expression in keratinocytes, further reinforcing the potential of AQP3-targeted treatments for aging skin [165]. However, uncontrolled AQP3 overexpression can enhance epidermal proliferation and may lead to hyperproliferative skin disorders such as psoriasis and skin tumors [95]. Liu SH et al. investigated the role of AQP8 in human dermal fibroblasts, demonstrating that AQP8 helps mitigate oxidative stress by facilitating the H_2_O_2_, thereby reducing ROS levels. This decrease in oxidative stress promotes the expression of critical proteins such as collagen type I alpha 1 chain (COL1A1) and keratin 19 (KRT19), which are vital for maintaining skin integrity and overall health. These findings suggest that upregulating AQP8 could serve as a valuable therapeutic approach to preventing oxidative damage and decelerating skin aging [50]. Combined with the therapeutic potential of AQP3, as seen through its regulation via PPARγ and related pathways, as well as the benefits of Aloe barbadensis extract and trimethylglycine, AQP-targeted treatments could be pivotal for improving skin hydration, reducing oxidative stress, and combating the effects of aging on the skin. Bioactive phytocompounds have been shown to effectively regulate the expression and function of AQPs, including those in the skin [121,166]. Recent research has focused on developing antibodies to adjust the AQP3 channel and identifying microRNAs that interfere with AQP3 expression, aiming to create targeted therapies with low toxicity using biological molecules [95]. Key challenges in miRNA application include validating predicted miRNAs, ensuring efficient regulation, assessing therapeutic effects in relevant cell models, and managing off-target effects. The small size of miRNAs increases the likelihood of unintended interactions with endogenous mRNAs, and hairpin RNA structures can produce miRNAs with opposing functions, complicating their therapeutic use [58]. The emergence of potent, isoform-specific synthetic AQP inhibitors also presents new therapeutic possibilities [160,167,168]. These innovative compounds offer encouraging options for treating various diseases and may also contribute to strategies for slowing skin aging. Since AQPs are constitutively expressed in the plasma membranes and intracellular organelles of various cells, investigating AQP modulators for topical drug delivery provides a low-toxicity alternative. This method maintains steady drug levels, improves adherence, and avoids gastrointestinal degradation and dosing errors [169]. Nanocarriers are also increasingly used in skin disease treatments, enhancing penetration and retention while minimizing systemic absorption and side effects [170].

## 8. Conclusions and Future Perspectives

In conclusion, the regulation of the levels and functions of APQs, particularly in the skin, holds substantial therapeutic potential for mitigating skin aging and related conditions. Bioactive phytocompounds, antibodies, and isoform-specific synthetic inhibitors targeting AQPs hold promise for developing effective, low-toxicity treatments. However, further validation of these modulators, along with the discovery of more selective molecules and advanced topical drug delivery systems, particularly for modifying AQP function in the skin, is crucial for advancing these therapies. A deeper understanding of the molecular mechanisms, including how signaling pathways, transcription factors, and post-translational modifications regulate AQP functions, is necessary. This will help identify new therapeutic targets to maintain skin hydration, improve barrier integrity, and address age-related changes in skin physiology. Continued efforts in this area could lead to advances that enhance skin resilience and also address broader challenges associated with oxidative stress, cellular aging, and tissue degeneration.

## Figures and Tables

**Figure 1 biology-13-00862-f001:**
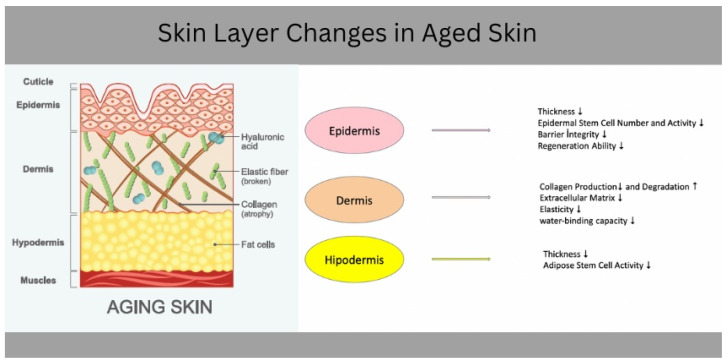
Changes in aged skin layers: This Figure highlights epidermal thinning, reduced dermal collagen, hypodermal fat loss, and stem cell depletion across various layers, leading to impaired barrier function and increased skin fragility.

**Table 1 biology-13-00862-t001:** Aquaporin roles in skin.

Aquaporin	Location in Skin	Functions	Reference
AQP1	Endothelial cells of dermal capillaries, melanocytes, fibroblasts	Water transport, cation channel activity, cell migration during wound healing	[2,75,76,77,78,79,80]
AQP3	Keratinocytes in stratum basale and stratum spinosum	Water, glycerol, and hydrogen peroxide transport, skin hydration, keratinocyte proliferation and differentiation, immune function, oxidative stress protection, circadian rhythm regulation	[74,81,82,83,84,85,86,87,88,89,90,91,92,93,94,95,96,97,98,99,100,101,102,103,104]
AQP5	Eccrine sweat glands	Keratinocyte proliferation, sweat secretion	[5,10,109,110]
AQP7	Hypodermis, dermal and epidermal dendritic cells	Antigen capture, migration of dendritic cells, immune response, allergy induction	[56,74,86,111,112]
AQP8	Dermal fibroblasts	Protection against oxidative stress (H_2_O_2_), prevention of aging-related damage	[50]
AQP9	Epidermal keratinocytes, neutrophils	Keratinocyte differentiation, immune response regulation, neutrophil function in immune reactions	[113,114,115,116,117,118,119]
AQP10	Stratum corneum of epidermis	Contributes to skin barrier function, lipid metabolism	[74,120]
AQP11	Hypodermal fat, adipocytes	Glycerol mobilization, triacylglycerol synthesis, lipid metabolism	[5,45,46]

**Table 2 biology-13-00862-t002:** Aquaporins and their involvement in the hallmarks of skin aging.

Hallmark of Aging	Aquaporin Involvement	Reference
Mitochondrial Dysfunction	AQPs (AQP1, AQP3, AQP5, AQP8, AQP9) facilitate H_2_O_2_ transport, regulating oxidative stress and mitochondrial dynamics. AQP3 modulates T cell migration and oxidative stress response; its expression declines in aging skin.	[120,121,122,123,124,125,126,127]
Cellular Senescence and Stem Cell Depletion	AQP3 overexpression improves human dermal fibroblast viability, inhibiting cellular senescence. AQP5 expression declines with age in epidermal stem cells, impairing tissue repair.	[10,59,138]
Impaired Macroautophagy	AQP3 interacts with autophagy proteins enhancing autophagy under short-term UVA exposure. Prolonged UVA exposure suppresses autophagy and accelerates aging.	[141,142]
Dysbiosis	Probiotics increase AQP3 expression, improving skin barrier function and reducing inflammation. This helps counteract dysbiosis effects on skin health.	[148,149]
Inflamm-Aging	AQP3 and AQP8 regulate oxidative stress and inflammation in skin cells. Reduced AQP3 levels lead to increased inflammation and skin aging, while AQP8 protects against oxidative stress.	[50,53,151,153,157]

Aquaporins (AQPs).

## Data Availability

Not applicable.

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
