# Peer review of "Aquaporin Channels in Skin Physiology and Aging Pathophysiology: Investigating Their Role in Skin Function and the Hallmarks of Aging"

_biology, 2024, doi:10.3390/biology13110862_

Round 1

Reviewer 1 Report

Comments and Suggestions for Authors

Vahid Ahmadi and Nazlı Karimi explores the role of aquaporins, proteins known for water transport, in skin health. Beyond hydration, aquaporins support barrier integrity and immune function by transporting essential molecules and activating key cellular pathways. There are several questions need to be addressed:

1. The authors claim that “understanding the mechanisms of age-related skin dysfunction could create the conditions for therapies that improve both skin health and also slow aging in other organs.” However, this statement requires stronger evidence. Further exploration of how compromised skin barriers can trigger inflammation and metabolic changes across multiple organs would solidify this connection.

2. A visual representation comparing young and aged skin would enhance the clarity of the section on skin structure and function. The figure should highlight changes like reduced collagen density, thinning of the epidermis, diminished elasticity, and loss of hydration. It would also be beneficial to summarize the role of epidermal and dermal stem cells, whose dysfunction or depletion contributes to the skin’s structural decline and impairs repair mechanisms. Including these elements will provide a more comprehensive understanding of how aging affects the skin's integrity.

3. In discussing the roles of aquaporins in skin function, it is essential to emphasize their contribution to the skin barrier’s ability to defend against fungal and bacterial infections. The discussion should also explore how AQPs are regulated at multiple levels: transcriptionally (involving NF-κB or HIF), translationally (via microRNAs), and post-translationally (through phosphorylation or glycosylation). These regulatory mechanisms are crucial for maintaining proper AQP function in healthy skin.

4. Since the skin serves as the body’s primary defense barrier, the hallmark signs of aging should be linked to weakened pathogen resistance. The inability to efficiently combat fungi and bacteria may contribute not only to localized infections but also to systemic inflammation, further exacerbating aging processes in other organs. This highlights the importance of maintaining skin health to protect overall physiological function.

5. Finally, potential off-target effects and toxicity concerns of AQP modulators must be discussed to ensure the safety of therapeutic interventions. The development of isoform-specific inhibitors remains a challenge, and these therapies must be carefully evaluated to avoid unintended side effects. Optimizing topical delivery systems will be essential to enhance bioavailability while minimizing systemic toxicity, ensuring effective and safe treatments for age-related skin conditions.

Reviewer 2 Report

Comments and Suggestions for Authors

Please add the source under the tables. Please add more research. 

Aim of the study: The aim of this study is to assess the role of aquaporins in skin physiology and in the aging process and their therapeutic potential. The strength of the work is primarily its topic, because today the aging processes of the skin play an important role in society, because aging is a deterioration of the function and quality of life.

General notes: the topic is very important, it touches on the role of aquaporins in the skin aging process and therapeutic processes. The authors reviewed the literature and reviewed aquaporin studies on both humans and mice. They indicate the role of these compounds in the functioning of the skin.

Specific remarks : Two tables were used in the paper. Under tables 1 and 2 one of them the source of the literature is not given - please add.

1. The review is clear, very comprehensively the authors show not only the role of aquaporins, but also dysbiosis and inflammatory aging of the skin. The article describes in detail the field it concerns.

2. It seems that the current review is still interesting, because the processes of skin aging along with lifestyle changes and the development of aesthetic medicine are and will be the subject of interest not only of scientists, but also and above all of the society, which currently cares a lot about its appearance. The authors are advised to look for more human research, which would be very valuable and would make the manuscript more interesting.

3. The bibliography consists of 140 items. These are not items from the last 5 years, they are also older, but extremely important in this issue and related to this issue.

4. The conclusions are consistent based on the literature used. It is only recommended to add other publications on human studies to the article. They already appear in the literature on the subject. In their summaries and conclusions, the authors emphasize the importance of the role of both the discussed cytokines, the microbiome, as well as inflammations in our skin and their role in therapeutic processes, but it has been correctly noted that the molecular basis needs to be better understood. 

Round 2

Reviewer 1 Report

Comments and Suggestions for Authors

The revision has incorporated the point I mentioned, making it well-suited for publication.